# Determinants of Caregiving Subgroups for Mexican American Caregivers Assisting Older Adults at Home and Their Influence on Perceived Stress

**DOI:** 10.3390/ijerph22091374

**Published:** 2025-08-31

**Authors:** Karen E. Schlag, Xiaoying Yu, Soham Al Snih, Monique R. Pappadis

**Affiliations:** 1Sealy Center on Aging, The University of Texas Medical Branch at Galveston (UTMB Health), Galveston, TX 77555, USA; soalsnih@utmb.edu (S.A.S.); mrpappad@utmb.edu (M.R.P.); 2Department of Population Health and Health Disparities, School of Public and Population Health, The University of Texas Medical Branch at Galveston (UTMB Health), Galveston, TX 77555, USA; 3Department of Biostatistics & Data Science, The University of Texas Medical Branch at Galveston (UTMB Health), Galveston, TX 77555, USA; xiyu@utmb.edu; 4Division of Geriatrics Medicine, Department of Internal Medicine, The University of Texas Medical Branch at Galveston (UTMB Health), Galveston, TX 77555, USA

**Keywords:** caregiving subgroups, Mexican Americans, latent class analysis, perceived stress

## Abstract

Patterns of family caregiving of older adults have been identified based on aspects such as care-related tasks and intensity and are associated with caregiver well-being. A gap remains, however, in understanding how individual-, relational-, and cultural-level factors concurrently inform caregiving groups within multicultural families. In this study, we identified caregiving patterns among Mexican American individuals aiding older adults by drawing from a variety of care recipient and caregiver characteristics. We also assessed relationships between established subgroups and perceived caregiver stress. Using data from the 2016 Hispanic Established Populations for the Epidemiological Study of the Elderly (Caregiver supplement, Wave 9, N = 460), we performed latent class analysis to determine caregiving subgroups from 8 indicator variables representing patient needs, family characteristics, and caregiver health and support. Findings identified four caregiving subgroups that varied based on older adults’ care needs and caregivers’ family status, nativity, and health. Results from multivariable linear regression indicated that caregivers from the *Moderate Burden/Non-cohabitating* group perceived significantly less stress than those in the *Elevated Burden & Health Risk* group. In conclusion, we demonstrated how multi-level factors shape caregiving patterns, which can inform support efforts for multicultural families.

## 1. Introduction

By 2040, the number of adults aged 65 and older in the United States is projected to reach 80 million, while individuals aged 85 and older are expected to grow to 14 million [1]. With more people living to advanced ages, there is a greater need for family members to care for older relatives desiring to age in place [2]. Older U.S. Hispanic adults make up a rapidly growing population, with projections suggesting they will comprise 21% of all older Americans by 2060 [3]. While U.S. Hispanic adults have frequently demonstrated longer life expectancies compared to other racial/ethnic groups, they have also been at high risk for comorbid health conditions and social and economic constraints later in life [4]. Furthermore, caregivers from Hispanic families are likely to provide a greater rate, duration, and intensity of care for their older relatives compared to individuals from non-Hispanic white families [5,6].

Mexican American adults represent the largest segment of older Hispanics in the U.S. [7] and are among minoritized groups facing healthcare access barriers and increased risk of delayed care and negative health outcomes compared to non-Hispanic white individuals [8,9]. Mexican American caregivers tend to be female, adult children, and experience greater financial strain from caregiving [6,10,11,12]. Adult children from this population, especially, may face greater stress when caring for parents born outside of the U.S., the latter of which may have higher dependency on their children compared to other sources of support [13]. Mexican American caregivers also tend to rely on family-based care over using formal support services [14], which can lead to increased care-related responsibilities. Notably, while use of formal support services for older adults has been associated with lower caregiving strain [15], this type of support has been related to poor health outcomes for Mexican American caregivers, potentially due to higher family stress when these services are implemented [14]. Notwithstanding, factors such as positive and strong family ties have been found to mitigate the adverse impact of caregiving stress on health outcomes for caregivers of Mexican descent [16].

### 1.1. From Theory to Caregiving as a Multidimensional Process

Prevailing frameworks including the stress process model [17] and the sociocultural stress and coping model [18] outline how various contextual factors (e.g., social determinants of health (SDOH), also known as non-medical drivers of health) and caregiving aspects (e.g., care recipient dependency) influence family caregivers’ experiences with stress and burden, which can adversely affect their health. Research drawing from these theories has shown, for instance, how factors such as the degree of a care recipient’s functional impairment or disruptive behavioral symptoms, which can increase caregiving demand, contribute to greater caregiver burden and distress across populations [10,19]. Family and individual characteristics, such as caregivers living with a care recipient, being an adult child, providing financial assistance, or having to manage their own health concerns, have also been related to increased caregiving stress [20,21,22,23]. Importantly, despite extensive efforts to identify separate influences on burden and health among family caregivers of older adults, a more holistic approach that examines patterns within caregivers’ heterogeneous experiences can offer nuanced insight into caregiving, including recognizing groups most vulnerable to stress [24].

Efforts to develop and categorize caregiving groups have provided theoretical and practical knowledge regarding how differences within individual caregiving domains contribute to caregiver outcomes. For instance, latent caregiver classes established based on differences in a family member’s caregiving intensity [25,26] have predicted variation in outcomes such as caregiver health, quality of life, and formal service use. Latent classes determined from different types of performed caregiving [27] or levels of resiliency [28] have also predicted caregiver variation in strain and well-being. Other caregiver domains used to classify caregiving groups include caregiver perceived competency [29], social support [23], care location [30], and caregiving management styles [31]. Notwithstanding this work, a gap remains in understanding how categories of caregiving can be informed by relationship- and cultural-level factors alongside individual care-related characteristics, especially within multicultural families.

### 1.2. Present Study

To the best of our knowledge, no research has examined how the varied experiences and characteristics of Mexican American caregivers who assist older adults contribute to distinct caregiving profiles as well as how group membership shapes caregiver well-being. Given this gap in the literature, we draw from the stress process [17] and sociocultural stress and coping models [18] as well as extant literature to consider how older adults’ functional and financial assistance needs along and their family caregiver’s health, living circumstances, nativity, status as an adult child, and formal support utilization inform unique categories of caregiving experiences. Additionally, we examine the extent to which the identified family caregiver subgroups among this population relate to their perceived stress. In doing so, our objective is to highlight how specific constellations of individual, family, and cultural factors contribute to the perceived stress of Mexican American caregivers of older adults.

## 2. Materials and Methods

### 2.1. Participants and Data Source

Participants were from the 2016 Hispanic Established Populations for the Epidemiological Study of the Elderly (H-EPESE) Wave 9 Caregiver supplement. The H-EPESE is an ongoing longitudinal cohort study initiated in 1993–1994 to assess the health and social determinants of health (SDOH) of Mexican Americans aged 65 and older residing in five U.S. states: Arizona, California, Colorado, New Mexico, and Texas. Data are collected via in-home cohort interviews occurring every 2–3 years in Spanish or English based on the respondent’s preference. Interviews collect data encompassing older Mexican Americans’ health conditions, physical and cognitive functional status, and SDOHs. Efforts to collect caregiver data began in 2010, wherein older adult survey respondents were asked to share contact information for a person they relied on most often for assistance and support. Named caregivers were interviewed about their perceptions of the health, functional status, and family and living circumstances of both the older care recipient and themselves.

For the current study, we analyzed responses from a sample of 460 caregivers interviewed in 2016, of which 61.3% completed their interview in Spanish. Data missingness was under 5% and most common on items reflecting caregiver perceived stress. For instance, the largest data missingness was for caregiver responses to a question regarding feeling unable to overcome difficulties (*n* = 14). Other items capturing caregiver perceived stress had similar missingness (*n* = 13), including feeling unable to control life, feeling confident, and feeling things were going their way.

### 2.2. Measures

***Indicators for caregiving profiles***. To identify caregiving profiles, we included 8 indicator variables. *Adult child caregiver* was measured based on participants’ indication of their relationship to the care recipient, which encompassed the following options: spouse, son/daughter, son/daughter-in-law, grandchild, brother/sister, nephew/niece, cousin, great-grandchild, other, friend, paid employee, sister/brother-in-law, or other relationship type. From these responses, this variable was dichotomized to indicate caregivers who were adult children of a care recipient or another relationship type. *Caregiver nativity* was assessed based on participants’ indicating they had been born in the U.S., in Mexico, or other. In consideration that only five caregivers selected “other” as their place of origin, they were excluded. The nativity variable was then dichotomized to reflect those born in the U.S. or Mexico. *Lives with care recipient* was coded dichotomously (yes/no). *Financial support* was measured as a caregiver indicating whether or not they provided financial assistance to a care recipient in general (e.g., bill paying, groceries) and/or for medical expenses (yes/no). Caregivers who responded yes to financially assisting their relative for either or both scenarios were coded as caregivers providing financial support, whereas caregivers responding no to both questions were coded as caregivers not providing financial support.

*Care recipient function* was determined based on summing how many out of 7 activities of daily living (ADLs) care recipients needed assistance with, including walking across a small room, bathing, grooming, dressing, eating, getting from a bed to a chair, and using a toilet. Using this information, this variable was dichotomized to reflect care recipients who needed help with two or more ADLs versus those needing assistance with only one ADL or no assistance. *Care recipient neuropsychiatric symptoms* were based on caregiver responses to the 12-item Neuropsychiatric Inventory (NPI) [32]. For this scale, caregivers indicated whether or not a care recipient had demonstrated any symptoms or changes in symptoms such as experiencing delusions or hallucinations, feeling anxious, irritable, excessive elation, or apathy, engaging in disruptive or repetitive behaviors, or demonstrating a change in appetite. Caregivers also rated the severity (i.e., mild, moderate, severe) for each NPI symptom they endorsed as present for the care recipient. For symptoms rated as none, caregivers skipped the severity question (both rated as zero). To create the indicator variable, we first multiplied each symptom frequency by its corresponding severity score and then summed these values across all symptoms. Then, we dichotomized this variable to reflect care recipients who fell below the median NPI score (i.e., less than 2 symptoms or changes in the past month) and those who were at the median or higher. We selected this cutoff point to help establish a relatively equal distribution of groups [33].

*Caregiver comorbidity* was determined from summing participants’ responses to individual questions indicating whether or not (yes/no) they had any of the following conditions: arthritis, cancer, heart attack, stroke, hypertension, diabetes, hip fracture, or bone fracture. We used this information to then dichotomize this variable as a caregiver having 2 or more of the listed conditions versus having one or fewer.

*Formal support* was measured based on caregivers’ responses to five questions assessing whether or not they used services in the past year, including respite care, caregiving training, help with care service acquisition, housing assistance, or medication management. It is notable that participants’ reliance on formal support overall was low, with frequencies reflecting caregivers’ individual service utilization ranging between 12.6 and 2 percent of the sample. As such, we summed caregivers’ responses to formal service questions (yes/no) and created a dichotomous variable reflecting caregivers who had used at least one service versus no services.

***Caregiver perceived stress.*** We used an abbreviated version of the Perceived Social Stress Scale (PSS) [34] to assess our outcome variable of caregiver perceived stress. To complete this measure, caregivers reported how frequently (0 = *never*, 4 = *very often*) they felt (1) unable to control important things in life; (2) confident in handling personal problems; (3) that things were going their way; and (4) that difficulties were piling up so high they could not overcome them. Items 2 and 3 were reverse-coded. We summed the four items, resulting in a range between 0 and 13, with higher scores representing greater perceived stress (*M* = 3.8, *SD* = 2.6).

***Covariates***. We considered caregiver sex (i.e., male, female), age in years, and education as covariates when testing the extent to which caregivers’ class membership into caregiving subgroups predicted their perceived stress. For education, we used participants’ reporting of years in school to create a categorical variable indicating they had (1) completed up to 8 years of school, (2) between 9 and 12 years, and (3) 13 or more years. We selected categories representing below, equivalent to, and above high school attainment, considering prior research linking education level and caregiving intensity among Mexican American caregivers [6].

### 2.3. Statistical Analysis

We used SPSS, version 29, for descriptive statistics (i.e., frequencies, means, standard deviations). To identify caregiver profiles, we conducted latent class analysis (LCA) using Stata, version 18. LCA discerns mutually exclusive categorical latent variables, maximizing heterogeneity between and homogeneity within identified classes [35]. We initiated this process by estimating a one-class model and then adding additional classes until identifying the best-fitting model [36]. To assess model fit, we drew in part from theoretical principles of caregiving outlined in the stress process [17] and sociocultural stress and coping models [18], as well as our knowledge of extant literature on family caregiving characteristics in Hispanic families. We also examined statistical indicators of model fitness, including the Akaike Information Criterion (AIC) [37] and Bayesian Information Criterion (BIC) [38], with lower values for both criteria suggesting a better balance between fit to the data and model parsimony [39]. Additionally, we assessed each model’s entropy score, which ranges from 0 to 1, as a diagnostic criterion, considering those having a score of at least 0.7 to indicate acceptable model classification accuracy or separation between classes [40]. We also examined the average latent class posterior probability for each class model, seeking those with scores greater than 0.7 to suggest individuals had been reliably assigned to a latent class based on their responses to indicator variables [41]. Finally, we examined each model’s individual class sizes to determine if any had classes containing less than 5% of the sample, which, while not a set rule for determining model fitness, suggests greater parameter estimate reliability and statistical power [42].

Next, to test the extent to which caregivers’ membership into a specific caregiving profile predicted their perceived stress, we conducted multivariable linear regression using SPSS version 29 while including covariates of interest. First, we conducted chi-square tests of independence and one-way ANOVA to examine relationships between caregiver profile class membership and potential covariates. Results indicated no significant relationship between caregivers’ profile membership and their age and years of education; however, we did find sex to be significantly associated with profile membership (X^2^ (3) = 67.610; *p* < 0.001). Consequently, we excluded caregivers’ sex as a covariate. To run the regression model, we entered caregivers’ age and years of education into Block 1, examining solely covariate effects on perceived stress, followed by adding profile class membership into Block 2, which considered the influence of all entered variables on stress. Additionally, we assessed potential multicollinearity in our regression model, with results indicating the variance inflation factor ranging between 1.08 and 1.55, with no evidence of collinearity. Residual diagnosis and Q-Q plot showed that assumptions of normality, linearity, and homoscedasticity were met.

## 3. Results

Table 1 provides descriptive statistics for the study sample and indicator variables used to identify caregiver subgroups of Mexican American adults caring for older relatives at home. A majority of the participants were female (77.6%) and had attended high school (40%), with the total sample having an average age of 59.6 years. A notable percentage of caregivers were also adult children (62%), from the U.S. (57.1%), and lived with the relative they cared for (51%). Table 1 offers additional indicator variable descriptives.

### 3.1. Identifying Caregiver Profiles Using Latent Class Analysis

LCA models with one to four classes were fit to identify caregiver subgroups for Mexican American adults. Table 2 shows model fit statistics for the different models tested.

While the BIC score suggested the most support for a two-class model, further examination of this statistic with AIC scores and entropy values, as well as consideration of existing literature on caregiving within Hispanic families, led us to select the four-class model. The AIC tends to favor more complex models than the BIC. It is common in LCA model testing to have inconsistent findings across fit indicators [39]. We also examined the average posterior probabilities for each class, with each of these being greater than 0.7. Finally, class sizes for each of the models were greater than 50 and 5% of the population. Figure 1 presents a graphic representation of the four-class model and the probabilities of each indicator variable in a specific subgroup. Probabilities closer to one shown on the y-axis indicate a higher likelihood of class membership for a particular indicator variable.

A majority of caregivers (32%) made up the elevated *Burden & Health Risk* profile. These caregivers had the lowest probability of being adult children compared to other subgroups and were highly likely to be Mexican-born. They were moderately likely to live with and provide financial support to a care recipient who had moderate-to-high functional assistance and support needs. Caregivers in this profile also had a moderate-to-high likelihood of having a comorbidity, though they were very unlikely to use formal support. The next largest caregiver profile (29%) was *High Burden/Low Health Risk*, with these participants highly likely to be adult children born in the U.S. who provided substantial financial support to a relative with greater functional assistance and support needs. These caregivers were also relatively likely to live with the care recipient while having a low-to-moderate probability of having a comorbidity. Similar to the other profiles, these caregivers were not likely to use formal support.

The third largest class (22%) was *Low Burden/Formal Supports*. Caregivers making up this profile were moderately likely to be an adult child caregiver born in the U.S., live with the care recipient, and provide financial support to an older adult who was likely to need only minimal functional assistance or support needs. Caregivers in this group also had a low-to-moderate probability of having a comorbidity. They were also the group most likely to use at least one type of formal assistance. The group with the smallest class size was *Moderate Burden/Non-cohabitating*. Caregivers in this profile had a high probability of being an adult child born in the U.S., while being highly unlikely to live with the care recipient or provide them with financial support. Older adults being cared for in this group were moderately likely to require functional assistance and support. Caregivers fitting this profile had a low-to-moderate chance of having comorbidity and were the least likely to use formal support assistance compared to the other classes.

### 3.2. Multivariable Linear Regression Model

Results from running multivariable linear regression using caregiver perceived stress as the dependent variable and caregiver age, education, and profile class membership as predictors are shown in Table 3.

The overall model with all three predictors was significant and explained 4% of the variance in caregiver perceived stress with *F*(6, 433) = 4.05, *p* < 0.001. While the adjusted R^2^ of the final model was low, the addition of the latent class membership variable along with age and education resulted in a significant increase in explained variance (ΔR^2^ = 0.035), suggesting that group membership contributed meaningfully to the prediction of caregiver perceived stress even after controlling for the other factors. Additionally, this final model showed a significant association between class membership and caregivers’ perceived stress. Specifically, holding age and education constant, caregivers comprising the *Moderate Burden/Non-cohabitating* class perceived significantly less stress than caregivers in the *Elevated Burden & Health Risk* class (standardized beta coefficient β = −0.16, *p* < 0.01), the latter used as the reference variable for class membership. Caregivers in the *High Burden/Low Health Risk* and *Low Burden/Formal Supports* subgroups were not significantly likely to perceive more or less stress compared to those in the *Elevated Burden & Health Risk* category (β = 0.10, p = 0.08; β = 0.05, *p* = 0.43, respectively). Examination of the covariates in model 2 showed that caregivers who attended 8 or less years of school perceived significantly more stress than caregivers who had completed 13 years of school or more (β = 0.14, *p* = 0.02). Caregivers’ age did not significantly predict their perceived stress (β = −0.03, *p* = 0.59).

## 4. Discussion

This study used care recipient and caregiver characteristics to develop latent caregiving classes for Mexican American caregivers of older adults. Specifically, we identified four distinct classes of caregiving, including *Elevated Burden & Health Risk*, *High Burden/ Low Health Risk*, *Low Burden/Formal Supports*, and *Moderate Burden/Non-Cohabiting.* In doing so, we highlighted differences in the caregiving experiences for members of this population based on the interplay of individual, relational, and cultural factors. Our findings also reinforced the notion that while the act of caregiving can function as a source of stress, caregivers may be better suited to cope when not overtaxed with additional external stressors.

The two largest caregiving subgroups in this study (i.e., *Elevated Burden & Health Risk* and *High Burden/Low Health Risk*) comprised a combined 61% of the sample and included Mexican American caregivers who were highly likely to assist older adults with substantial functional limitations and disruptive behavioral symptoms, while also providing these care recipients with financial assistance. These two classes reflect the increased risk of chronic illnesses such as Alzheimer’s disease, diabetes, and hypertension among older Hispanic Americans who primarily rely on family caregivers for their physical and emotional support needs [11,43]. Notably, older care recipients in these two classes had a moderate-to-high likelihood of experiencing both functional limitations as well as neuropsychiatric symptoms. Moreover, our finding that caregivers assisting older adults with substantial physical and psychological support needs were also the most likely to provide them with financial support underscores the overall caregiving burden characterizing these two subgroups. Additionally, the finding that most caregivers in this study were moderately to highly likely to provide financial support to older care recipients reinforces an association between shared family economic resources and the cultural norm of filial obligation among Hispanic families [44].

When examining differences among class characteristics, the greatest variation was seen among the indicator variables of financial support and nativity. Compared to other groups, caregivers in the *High Burden/Low Health Risk* class were the most likely to provide financial assistance to older care recipients who had the highest probabilities of needing support for functional limitations and behavioral symptoms. Caregivers in the *Moderate Burden/Non-cohabitating* class were the least likely (less than one percent probability) to provide financial support to an older care recipient compared to other classes, although they were still likely to assist a relative with moderate functional limitations and disruptive symptoms. Regarding nativity, caregivers in the *High Burden/Low Health Risk* class also had the highest likelihood of being U.S.-born as well as an adult child, compared to other groups, while those in the *Elevated Burden & Health Risk* class were the most likely to be Mexico-born and not an adult child.

The indicator variable of formal support assistance had the greatest similarity in probabilities across the four caregiving classes. In general, caregivers reported having a low probability of using at least one type of formal support service, which is consistent with extant research [13,45]. Mexican American caregivers may be less prone to use formal support resources in part due to dissatisfaction with existing services, cultural preferences for family-based care, and higher family stress when these services are implemented [14,45,46]. Interestingly, the *Low Burden/Formal Supports* class, where care recipients were the most likely to have minimal functional and behavioral support needs, was the one subgroup wherein caregivers indicated being moderately likely to use formal support services such as respite care or assistance with locating support services. Findings point to the need for research to identify formal support service implementation tailored to more effectively address the needs of this population [13].

Indicator variables of caregiver health and living with a care recipient had notable patterns across caregiving classes. For example, caregivers in three of the identified classes (64% of the sample) had a low-to-moderate probability of having a comorbidity. Only caregivers in the *Elevated Burden & Health Risk* group reported a higher comorbidity likelihood, with these individuals also more likely to be Mexico-born. While some research has found foreign-born caregivers in the U.S. to report better or no difference in health outcomes compared to their U.S.-born counterparts [47], other studies have indicated that immigration-related stressors, such as discrimination and acculturation, may contribute to physiological strain and health decline over time [48,49]. Most caregivers in our study (i.e., belonging to the three largest caregiving classes and making up 83% of the sample) also had a moderate-to-high probability of cohabitating with the relative they supported, which is consistent with extant research [6].

Regarding perceived stress, caregivers in the *Moderate Burden/Non-cohabitating* class experienced significantly lower stress than those in the *Elevated Burden & Health Risk* subgroup. In line with stress process frameworks [17,18] and extant studies linking caregiving intensity and tasks to caregiver distress [27,50], participants in our study who perceived significantly higher stress (i.e., *Elevated Burden & Health Risk class*) also had a greater probability of assisting a relative with substantial support needs, compared to those in the *Moderate Burden/Non-cohabitating* subgroup. However, given that perceived stress did not significantly differ between the *Moderate Burden/Non-cohabitating* and the *High Burden/Low Health Risk* subgroups (whose care recipients were even more likely to exhibit pronounced support needs), our findings point to factors beyond direct caregiving and older care recipient support needs as potential drivers of stress for Mexican American caregivers. Future research should explore additional influences on perceived caregiver stress, such as positive aspects of caregiving (PAC). PAC, including feeling appreciated and finding meaning as a caregiver, has been associated with outcomes such as reduced caregiving burden [51,52]. Additionally, Hispanic caregivers have reported higher scores in PAC compared to other racial or ethnic groups [53], suggesting these attitudes may especially contribute to lower perceived stress for this population, including among subgroups facing high caregiving intensity.

### 4.1. Implications

Our study underscores the need for interventions to address caregiver support needs beyond direct caregiving, such as tailored programs focusing on family caregiver physical health. Findings from our research also inform future studies and policymaking by emphasizing the need to design formal support services that are more aligned with the needs and values of families who have historically faced barriers to assistance programs. Finally, future studies can benefit from considering differences in caregiver support needs based on whether they live with or apart from older care recipients.

### 4.2. Limitations

While this study contributes unique knowledge about caregiving classes among Mexican American caregivers of older adults and how subgroups predict caregiver stress, findings should be considered alongside several limitations. First, findings indicated that the explained variance of our regression model to test predictors of caregivers’ perceived stress was low (adjusted R^2^ = 0.04). Notably, however, the addition of caregiving class membership into our model resulted in a significant increase in explained variance (ΔR^2^ = 0.035, *p* < 0.001), indicating that our primary outcome variable added meaningfully to caregivers’ perceived stress. Secondly, while research has consistently recognized values of familism as mitigating perceived caregiving burden within Hispanic populations [12], we were unable to consider family identification or loyalty as an influence on caregiving subgroups or stress, as our sample displayed minimal variation in attitudes of familism (i.e., greater than 80% of caregiver responses indicated agreeing or strongly agreeing with questions related to familism; e.g., *Married children should live close to their parents so they can help each other*). Future research should consider how family and cultural norms play a role in the care provision of older Mexican American caregivers. Third, considering our focus on Mexican American caregivers living in either Texas, New Mexico, Colorado, Arizona, and California, we are unable to extend findings from this study to other caregiving groups or those in other areas of the U.S., such as Illinois, where 79% of Hispanics identify as Mexican [54]. Finally, the data used in this study are from 2016, which may not reflect the current state of older Mexican American caregivers in the U.S. Future research should utilize prospective or qualitative study designs to pursue more nuanced understandings of various influences on caregiver stress implied from our findings. Subsequent studies can also benefit from exploring how caregiving profiles and health risks change with time.

## 5. Conclusions

In this epidemiological cohort study including older Mexican American caregivers, four distinct caregiving subgroups were identified based on caregiver characteristics, formal support utilization, relationship to the older care recipient, and the neuropsychiatric symptoms and functional limitations of the care recipient. These distinct caregiving categories also influenced caregivers’ perceived stress. Adult family caregivers with elevated health risk and born outside of the U.S. who are caring for their loved ones with significant functional needs tend to report greater perceived stress than those with less care-related burden and better health. Future work should explore the dynamic multi-level factors that influence caregiving patterns over time to inform support provision for multicultural families.

## Figures and Tables

**Figure 1 ijerph-22-01374-f001:**
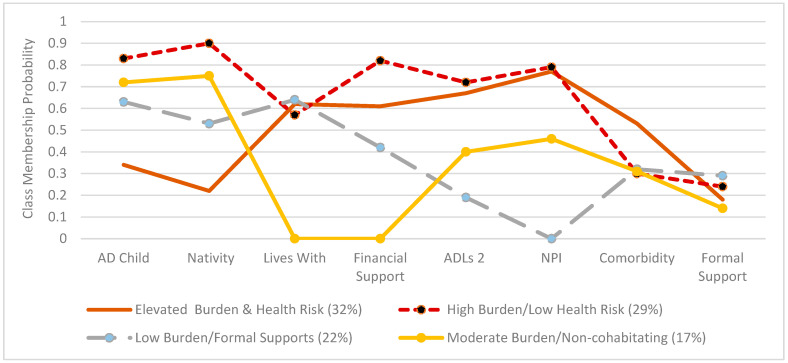
Probability of indicators for each class. Responses closer to one on the y-axis represent greater caregiver endorsement of a particular trait. AD Child CG = adult child caregiver; Nativity = closer to one represents higher probability of a caregiver U.S.-born; ADLS 2 = care recipient needs assistance with at least two activities of daily living; NPI = care recipient experiences at least 2 neuropsychiatric symptoms; Comorbidity = caregiver has a comorbidity; Formal Support = caregiver uses at least one formal support service to assist with care of an older relative.

**Table 1 ijerph-22-01374-t001:** Caregiver sample and subgroup indicator characteristics.

Caregiver Characteristics	*n* (%)	Mean (SD)
Age		59.6 (13.2)
Sex		
Female	357 (77.6)	
Male	103 (22.4)	
Education, years completed		
1–8 years	134 (29.1)	
9–12 years	184 (40.0)	
13+ years	142 (30.9)	
**Indicator variables**		
Adult child caregiver		
Yes	285 (62)	
No	175 (38)	
CG nativity		
U.S. born	260 (57.1)	
Mexican born	195 (42.9)	
CG lives with CR		
Yes	234 (51)	
No	225 (49)	
CG provides financial support to CR		
Yes	239 (52.6)	
No	215 (47.4)	
CR needs assistance with at least 2 ADLs		
Yes	247 (53.7)	
No	213 (46.3)	
CR has at least 2 neuropsychiatric symptoms		
Yes	257 (55.9)	
No	203 (44.1)	
CG has a comorbidity		
Yes	172 (38.3)	
No	277 (61.7)	
CG uses at least one caregiving formal service		
Yes	99 (21.7)	
No	357 (78.3)	

CG = caregiver; CR = care recipient; ADL = activities of daily living.

**Table 2 ijerph-22-01374-t002:** Class solutions.

Model Fit and Diagnostic Criteria							
Models	LL	AIC	BIC	Entropy	ALCPP	Smallest Class Size (n)	Smallest Class Size (%)
Class 1	−2419.09	4854.18	4887.23	--	--	460	100
Class 2	−2386.86	4807.72	4877.95	0.466	0.84	175	38
Class 3	−2368.99	4789.99	4897.40	0.459	0.76	137	29.8
Class 4	−2353.75	4777.49	4922.09	0.600	0.77	99	21.5

*N* = 460. The model became unstable upon testing a 5-class model. LL = log-likelihood; AIC = Akaike information criterion; BIC = Bayesian information criterion; ALCPP = average latent class posterior probability.

**Table 3 ijerph-22-01374-t003:** Profile class membership related to caregiver perceived stress.

	Model 1				Model 2			
Predictors	B	S.E.	β	*p*	B	SE	β	*p*
**Education 1**	**0.71**	**0.33**	**0.12**	**0.03**	**0.81**	**0.34**	**0.14**	**0.02**
Education 2	−0.14	0.29	<0.01	0.65	−0.13	0.29	−0.02	0.66
Age	<0.01	0.01	−0.02	0.68	−0.01	0.01	−0.03	0.59
Profile Membership								
High Burden/ Low Health Risk					0.27	0.34	0.05	0.43
Low Burden/ Formal Supports					−0.62	0.35	−0.10	0.08
**Moderate Burden/** **Non-cohabitating**					**−0.98**	**0.34**	**−0.16**	**<0.01**
R^2^	0.011				0.040			
R^2^ change	0.018				0.035			

B = unstandardized coefficient; S.E. = standard error; β = standardized coefficient; Education 1 = completed 8 or less years of school; Education 2 = completed between 9 and 12 years of school. Education 3, representing completion of 13 or more years of school, is the reference variable for this covariate. *Elevated Burden & Health Risk* is the reference class for the LCA model. R^2^ = adjusted R^2^. Bold text indicates significant predictors.

## Data Availability

The original data presented in the study are openly available at the Hispanic Established Populations for the Epidemiologic Study of the Elderly (HEPESE) Wave 9, 2016 [Arizona, California, Colorado, New Mexico, and Texas] ICPSR 39038 at https://doi.org/10.3886/ICPSR39038.v2 (accessed 21 August 2025).

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
