# Peer review of "Determinants of Caregiving Subgroups for Mexican American Caregivers Assisting Older Adults at Home and Their Influence on Perceived Stress"

_ijerph, 2025, doi:10.3390/ijerph22091374_

Round 1
Reviewer 1 Report
Comments and Suggestions for Authors
This study adds significantly to the body of knowledge on family caregiving in Mexican American communities and is well-conducted and well-written. It is both methodologically sound and conceptually based on well-established theoretical frameworks like the stress process model and the sociocultural stress and coping model to use latent class analysis (LCA) to identify distinct caregiving subgroups based on multi-level factors—individual, relational, and cultural.
Given the ageing Hispanic population in the United States and their growing reliance on informal caregiving, the research question is pertinent and timely. H-EPESE is a reliable, nationally representative data source that is especially suitable for this demographic.
The authors' clear and convincing justification for choosing the four-class solution, which includes a discussion of model fit indices (AIC, BIC, entropy, and posterior probabilities), and the sample size (n = 460) are sufficient for LCA, which enhances the validity of the results.
The four caregiving profiles that have been identified—Moderate Burden/Non-cohabitating, Low Burden/Formal Supports, High Burden/Low Health Risk, and Elevated Burden & Health Risk—are distinct from one another and represent significant trends in caregiving experiences. The typology gains depth and transcends basic task-based classifications with the incorporation of nativity, cohabitation status, financial support, formal service use, and carer health. This is in line with the study's objective of documenting the complexity of caregiving in multicultural households.
Caretakers in the Moderate Burden/Non-cohabitating group reported significantly lower perceived stress than those in the Elevated Burden & Health Risk group, according to your multivariable regression analysis, which appropriately controls for important covariates. This demonstrates how outside stressors (such as carer comorbidity, foreign birth, or a lack of official support) can make caregiving more difficult than it would otherwise be. In light of structural barriers (like healthcare access), cultural norms (like familialism), and potential protective factors (like the positive aspects of caregiving), the discussion carefully interprets these findings.
The modest explained variance in the regression model, the low use of formal services limiting variability, and the data's geographic and temporal specificity are among the limitations that are truthfully and appropriately acknowledged. These don't call into question the study's contributions but instead offer clear directions for future research.
Minor suggestions for improvement :
1) Consider briefly discussing how the Positive Aspects of Caregiving (PAC) might interact with class membership, even if not measured here, as it could help explain why some high-burden groups (e.g., High Burden/Low Health Risk) did not report elevated stress.
2) In Figure 1, while the probabilities are well-presented, labelling the y-axis explicitly as “Probability” rather than relying solely on context would enhance clarity for readers.
3) A brief sentence in the methods could clarify why education was categorised into three groups (≤8, 9–12, ≥13 years), especially since this reflects below, at, and above high school completion, which may be meaningful in this population.
Overall, this manuscript is strong in design, execution, and presentation. It advances our understanding of caregiving heterogeneity in a rapidly growing and underserved population and offers actionable insights for tailored interventions.
Author Response
Comment 1: This study adds significantly to the body of knowledge on family caregiving in Mexican American communities and is well-conducted and well-written. It is both methodologically sound and conceptually based on well-established theoretical frameworks like the stress process model and the sociocultural stress and coping model to use latent class analysis (LCA) to identify distinct caregiving subgroups based on multi-level factors—individual, relational, and cultural.
Given the ageing Hispanic population in the United States and their growing reliance on informal caregiving, the research question is pertinent and timely. H-EPESE is a reliable, nationally representative data source that is especially suitable for this demographic.
The authors' clear and convincing justification for choosing the four-class solution, which includes a discussion of model fit indices (AIC, BIC, entropy, and posterior probabilities), and the sample size (n = 460) are sufficient for LCA, which enhances the validity of the results.
The four caregiving profiles that have been identified—Moderate Burden/Non-cohabitating, Low Burden/Formal Supports, High Burden/Low Health Risk, and Elevated Burden & Health Risk—are distinct from one another and represent significant trends in caregiving experiences. The typology gains depth and transcends basic task-based classifications with the incorporation of nativity, cohabitation status, financial support, formal service use, and carer health. This is in line with the study's objective of documenting the complexity of caregiving in multicultural households.
Caretakers in the Moderate Burden/Non-cohabitating group reported significantly lower perceived stress than those in the Elevated Burden & Health Risk group, according to your multivariable regression analysis, which appropriately controls for important covariates. This demonstrates how outside stressors (such as carer comorbidity, foreign birth, or a lack of official support) can make caregiving more difficult than it would otherwise be. In light of structural barriers (like healthcare access), cultural norms (like familialism), and potential protective factors (like the positive aspects of caregiving), the discussion carefully interprets these findings.
The modest explained variance in the regression model, the low use of formal services limiting variability, and the data's geographic and temporal specificity are among the limitations that are truthfully and appropriately acknowledged. These don't call into question the study's contributions but instead offer clear directions for future research.
Response 1: Thank you for this feedback! We are greatly appreciative!
Comment 2: Minor suggestions for improvement :
1) Consider briefly discussing how the Positive Aspects of Caregiving (PAC) might interact with class membership, even if not measured here, as it could help explain why some high-burden groups (e.g., High Burden/Low Health Risk) did not report elevated stress.
Response 2: We appreciate this point. We have elaborated on our mention of PAC in the discussion section to better convey the possibility (and suggest the need for future research) that PAC may function to lower Hispanic caregivers’ perceived stress, including among subgroups who face high caregiving intensity. The edited version (Lines 383-390) reads:
“Future research should explore additional influences on perceived caregiver stress, including positive aspects of caregiving (PAC). PAC, such as feeling appreciated and finding meaning as a caregiver, have been associated with outcomes such as reduced caregiving burden. [51, 52] Additionally, Hispanic caregivers have reported higher scores in PAC compared to other racial or ethnic groups, [53] suggesting these attitudes may especially contribute to lower perceived stress for this population, including among subgroups facing high caregiving intensity.”
Comment 3: 2) In Figure 1, while the probabilities are well-presented, labelling the y-axis explicitly as “Probability” rather than relying solely on context would enhance clarity for readers.
Response 3: Thank you for this suggestion! We have added a title for the y-axis titled “Class Membership Probability”. (pg. 7)
Comment 4: 3) A brief sentence in the methods could clarify why education was categorised into three groups (≤8, 9–12, ≥13 years), especially since this reflects below, at, and above high school completion, which may be meaningful in this population.
Response 4: We have added a sentence to the Methods section regarding our selection of the education categories (line 185-188). It reads: “We selected categories representing below, equivalent to, and above high school attainment considering prior research linking education level and caregiving intensity among Mexican American caregivers. [6]”
Comment 5: Overall, this manuscript is strong in design, execution, and presentation. It advances our understanding of caregiving heterogeneity in a rapidly growing and underserved population and offers actionable insights for tailored interventions.
Response 5: Thank you for this feedback!
Reviewer 2 Report
Comments and Suggestions for Authors
The subject matter the authors are tying to tackle is incredibly important and equally difficult. I am not entirely certain this is work that can be carried out via. retrospective analysis but the authors have been very methodical in their approach and analysis and their work does progress discussion in this area.
I would suggest the authors avoid drawing inferences in the discussion that are not supported by their data e.g. "may have faced additional stressors outside of caregiving that were related to their own health issues as well as discriminatory barriers to healthcare coverage and services (380 - 382). "...positive aspects of caregiving (PAC), such as feeling appreciated and finding meaning as a caregiver" (385-386).
While the authors indicate the limitations of the collection of the data in 2016, I may also include a discussion on the limitations of applying a retrospective cohort study design. The findings of this work suggest that there are caregiver characteristics that may impact caregivers' perceived stress. A prospective cohort or qualitative study design would allow for much more nuanced understanding of drivers (e.g. identification of future work). It is clear this group would be well positioned to further contribute to this meaningful work.
Author Response
Comment 1: The subject matter the authors are tying to tackle is incredibly important and equally difficult. I am not entirely certain this is work that can be carried out via. retrospective analysis but the authors have been very methodical in their approach and analysis and their work does progress discussion in this area.
Response 1: Thank you for this feedback!
Comment 2: I would suggest the authors avoid drawing inferences in the discussion that are not supported by their data e.g. "may have faced additional stressors outside of caregiving that were related to their own health issues as well as discriminatory barriers to healthcare coverage and services (380 - 382). "...positive aspects of caregiving (PAC), such as feeling appreciated and finding meaning as a caregiver" (385-386).
Response 2: We appreciate this point. To improve the clarity and accuracy of our discussion section, we have made edits to our original intended reflection that caregiver stress in our study was potentially influenced by factors outside of direct caregiving such that this sentence is now concise and based strictly from our findings. The edits (lines 382-383) read as:
“… our findings point to factors beyond direct caregiving and older care recipient support needs as potential drivers of stress for Mexican American caregivers.”
We also edited our discussion of PAC to help elaborate on an implication from our study (i.e., for future research to consider influences on caregiving stress beyond direct caregiving for this population). This choice was also in response to a suggestion from the other reviewer to consider how PAC might influence caregiver subgroups. Our edited version reads (lines 383-390):
“Future research should explore additional influences on perceived caregiver stress, including positive aspects of caregiving (PAC). PAC, such as feeling appreciated and finding meaning as a caregiver, have been associated with outcomes such as reduced caregiving burden. [51, 52] Additionally, Hispanic caregivers have reported higher scores in PAC compared to other racial or ethnic groups, [53] suggesting these attitudes may especially contribute to lower perceived stress for this population, including among subgroups facing high caregiving intensity.”
Comment 3: While the authors indicate the limitations of the collection of the data in 2016, I may also include a discussion on the limitations of applying a retrospective cohort study design. The findings of this work suggest that there are caregiver characteristics that may impact caregivers' perceived stress. A prospective cohort or qualitative study design would allow for much more nuanced understanding of drivers (e.g. identification of future work). It is clear this group would be well positioned to further contribute to this meaningful work.
Response 3: Thank you for this suggestion! We have added the suggested limitation as follows (lines 419-421):
“Future research should utilize prospective or qualitative study designs to pursue more nuanced understandings of various influences on caregiver stress implied from our findings.”